# NETWORK ITERATIVE LEARNING FOR DYNAMIC DEEP NEURAL NETWORKS VIA MORPHISM

## ABSTRACT

In this research, we present a novel learning scheme called *network iterative learning* for deep neural networks. Different from traditional optimization algorithms that usually optimize directly on a static objective function, we propose in this work to optimize a dynamic objective function in an iterative fashion capable of adapting its function form when being optimized. The optimization is implemented as a series of intermediate neural net functions that is able to dynamically grow into the targeted neural net objective function. This is done via network morphism so that the network knowledge is fully preserved with each network growth. Experimental results demonstrate that the proposed network iterative learning scheme is able to significantly alleviate the degradation problem. Its effectiveness is verified on diverse benchmark datasets.

## 1 INTRODUCTION

Deep convolutional neural networks have recently demonstrated their continuous excellent performances on diverse computer vision problems, including image classification (Lang et al., 1990; Simonyan & Zisserman, 2014; Szegedy et al., 2014; He et al., 2015a), object detection (Girshick et al., 2014; Ren et al., 2015), and semantic segmentation (Long et al., 2015). Deep neural network algorithms involve many optimization problems. One of the most important is network training. It is quite common to invest days to months of time on hundreds of machines in order to solve even a single instance of the neural network learning problem (Goodfellow et al., 2016).

The optimization of deep neural networks is much more difficult than traditional optimization problems. One of the most obvious difficulties is that the problem of optimizing a deep neural network is ill-conditioned (non-convex) and contains numerous local minima (Goodfellow et al., 2016). Using the fact that neural network parameters are symmetric, one can show that an $m$-layer neural network with $n$ units at each layer can impose $n!^m$ ways of hidden units arrangements. This indicates that there can be an extremely large or even uncountably infinite amount of local minima in a deep neural network optimization problem.

Local minima can be problematic if they have high cost in comparison to the global minimum. It remains an open question whether there are many local minima of high cost for networks of practical interest (Goodfellow et al., 2016). It is a common sense that deep neural network optimization algorithms will not converge to a global minimum but instead a local one. However, many practitioners start to believe that, for sufficiently large neural networks, most local minima have a low cost function value (Saxe et al., 2013; Dauphin et al., 2014; Goodfellow et al., 2014; Choromanska et al., 2015).

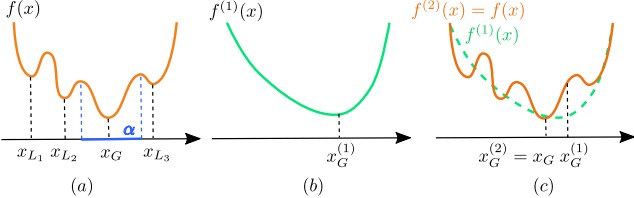

Figure 1: Network iterative learning intuition. Instead of (a) directly optimizing the objective function $f(x)$. We first (b) optimize a simpler approximate function $f^{(1)}(x)$, then (c) use its minimum $x_G^{(1)}$ as an initialization to further optimize $f^{(2)}(x) = f(x)$.

In this research, we shall show that in fact there is still a relatively large gap between the local minima and its global minimum for a large deep neural network optimized by traditional optimization algorithms. In order to further reduce this gap, we shall propose a novel network iterative learning

scheme. The intuition for the proposed network iterative learning scheme is illustrated in Fig. 1. Let $x_G$ be the global minimum of the function $f(x)$, and $x_{L_i}$, $i = 1, 2, 3$ be its local minima. Assuming that we use an uniform initialization, then a gradient-based optimization algorithm will have a probability of $\alpha$ ($0 < \alpha < 1$) to converge to the global minimum. Instead of directly optimizing the function $f(x)$, we first use a simpler function $f^{(1)}(x)$ to approximate $f(x)$, and let $f^{(2)}(x) = f(x)$. Using a gradient-based optimization algorithm, regardless of the initialization, the algorithm will converge to the global minimum of $f^{(1)}(x)$ at $x_G^{(1)}$. Suppose that $x_G^{(1)}$ falls into the convex subregion of $f^{(2)}(x)$ which contains $x_G$, a second round of a gradient-based optimization algorithm will converge to the global minimum of $f(x)$ at $x_G$. In this case, we will have a probability of 1 to converge to the global minimum of $f(x)$ using the proposed network iterative learning scheme. For complex functions such as deep neural networks, function $f^{(1)}(x)$ might not be convex and $x_G^{(1)}$ might not fall into the convex subregion of $f^{(2)}(x)$ containing $x_G$ either. We believe that the proposed network iterative learning scheme will have higher probability to fall into low cost local minima than directly optimizing the objective function $f(x)$. This belief was verified by the proposed experiments.

It is worth noting that the problem of local minima with high cost can not be rectified by gradient-based optimization algorithms, including the backpropagation-based algorithms for training deep neural networks. Theoretically, a local minimum is guaranteed to be a global minimum if the objective function is convex. However, for a deep neural network, its objective function is obviously non-convex (Bishop, 2006). Empirically, when we use different optimization algorithms to train the deep neural networks, they all converge to a similar performance, but can not converge to lower cost local minima comparing against the proposed network iterative learning scheme (Fig. 9).

The proposed network iterative learning scheme for deep neural networks is based on network morphism (Wei et al., 2016). Network morphism is an effective scheme to morph a well-trained neural network into a new one with the network function preserved. After morphing a parent network, the child network is expected to inherit the knowledge from its parent network and also has the potential to continue growing into a more powerful one. The idea of the proposed network iterative learning scheme for deep neural networks based on network morphism is illustrated in Fig. 4. The results are shown in Fig. 2(e). As shown, an 8-layer network is first learned, and then iteratively grown into 14, 26, 50, and 98 layers to achieve continuously improved performance[1].

The proposed learning scheme is not only a novel learning scheme, but also contains a novel deep dynamic neural network architecture. To achieve this, we introduce a unified network architecture called DynamicNet to enable the dynamic growth of neural networks. As illustrated in Fig. 3, this architecture follows a modularized architecture design. For each level, a repeatable template block is used to enable the dynamic change of its depth. DynamicNet can represent a large family of networks. In this research, we verify the proposed scheme on two classic architectures, i.e., PlainNet and ResNet (He et al., 2015a). Since these two networks are not initially proposed with architecture change, certain modifications need to be made to adapt them to the DynamicNet family. Furthermore, we also elaborate how each of these two networks can be iteratively evolved by the proposed learning scheme. The detailed network iteration processes are illustrated in Fig. 4. In this research, we also developed an algorithm based on optimal gradient descent to solve for the network morphism equation.

The proposed network iterative learning scheme is also capable of learning new knowledge based on the established knowledge base. This is because it avoids re-learning from scratch when the network is going deeper, but inherits the full knowledge that has already learned. It is known that when deeper neural networks are trained, a degradation problem may be encountered. As shown by the blue curve in Fig. 2(c), the error rate becomes saturated and then rapidly increases as the network depth increases. Such performance degradation is not caused by over-fitting (He et al., 2015a), because the training error of the deeper network is also larger than that of the shallower network (Fig. 2(a)). The degradation problem can be greatly alleviated by the proposed learning scheme. Its effectiveness is shown in Fig. 2(b), where the deeper network could achieve lower errors in both training and testing than the shallower one. Fig. 2(c) and (d) also compare these two learning schemes on PlainNet and ResNet, in which the performance improvement brought by the proposed scheme is indicated by the purple-pink region. In the proposed experiments, we shall also demonstrate that the models obtained

---

[1]Note that in Fig. 2(e), the sharp error increase and decrease at 0/160 epochs are caused by the learning rate change.

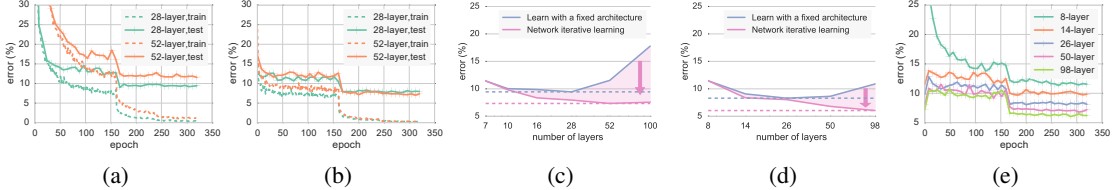

Figure 2: Network iterative learning vs. learning with a fixed architecture. (a) The degradation problem, i.e., the deeper network produces higher errors in both training and testing. (b) The degradation problem is overcome by the proposed learning scheme. (c,d) Comparison results of the two learning schemes on PlainNet and ResNet. (e) Testing error curves using network iterative learning. (a-c): PlainNet on CIFAR10, (d,e): ResNet on CIFAR10 (residual ratio 0.5).

by the proposed learning scheme are able to consistently achieve better performances than those learned with a fixed architecture on diverse network architectures.

## 2 RELATED WORK

*Local Minima of Deep Neural Network*    The object function of a deep neural network is non-convex. Hence, a local minimum can not be guaranteed to be also a global minimum (Bishop, 2006). There can be an extremely large or even uncountably infinite amount of local minima of a deep neural network optimization problem. Local minima can be problematic if they have high cost in comparison to the global minimum. It remains an open question whether there are many local minima of high cost for networks of practical interest (Goodfellow et al., 2016). Researchers have constructed small neural networks that have local minima with higher cost than the global minimum (Sontag & Sussmann, 1989; Brady et al., 1989; Gori & Tesi, 1992). In this research, we shall show that the problem of local minima with high cost in optimization is common and arises naturally for deep neural networks.

*Optimization Algorithms*    The main stream optimization algorithms for deep neural networks are gradient-based. They are mathematically derived from the chain-rule of calculating the derivatives of the composition of two or more functions in Calculus. Currently widely adopted gradient-based optimization algorithms include stochastic gradient descent (SGD), Nesterov's accelerated gradient descent (NAG) (Nesterov, 1983; 2013), RMSprop (Tieleman & Hinton, 2012), Adam (Kingma & Ba, 2014), AdaGrad (Duchi et al., 2011), AdaDelta (Zeiler, 2012), etc. However, the problem of local minima with high cost can not be rectified by these traiditonal gradient-based optimization algorithms. While the proposed network iterative learning scheme is able to further reduce the gap between local minima and the global minimum for deep neural networks.

*Knowledge Transferring and Network Morphism*    A series of work uses a student network to mimic the output of a teacher network (Bucilu et al., 2006; Ba & Caruana, 2014; Romero et al., 2014). Usually the student network is lighter and the teacher network is more complex. In such a teacher-student formula, the student network is usually relearned from scratch, and its network architecture is fixed. Another series of work (Chen et al., 2015a; Wei et al., 2016) tries to add new layers into the original network architecture, allowing the network architecture to dynamically change. Since the child network usually directly inherits the full knowledge of the parent network, we refer to this as the parent-child formula. The proposed network iterative learning is based on the latter series.

Network morphism (Wei et al., 2016) is an effective scheme to morph a well-trained neural network into a new one with the network function preserved. We choose it as the basic operation of the proposed network iterative learning scheme. However, this research is different from (Wei et al., 2016) significantly. One essential difference between the proposed research and (Wei et al., 2016) is that network morphism is used as a building block of the proposed network iterative learning scheme. Another essential difference is that, although only in theory, if we can add a single layer into a neural network, we are able to add an arbitrary number of layers. In practice, this might not be true. First, the choice of the morphing target is a great issue. Second, we need to guarantee that the performance will continuously improve when adding more layers. Otherwise, it would make no sense to do this. Surely, there will be a limit for the number of layers we are allowed to add due to the over-fitting problem. This paper presents a systematic study on how to pick the morphing target and how can we practically successful to morph a network by adding a large amount of layers, in which (Wei et al., 2016) did not address.

# 3  THE NETWORK ITERATIVE LEARNING APPROACH

## 3.1  NETWORK ITERATIVE LEARNING

Different from traditional optimization algorithms that usually optimize directly on a static objective function, we propose in this work to optimize a dynamic objective function in an iterative fashion capable of adapting its function form when being optimized.

In this research, we split such process into multiple stages, *with the knowledge learned in the previous stage being transferred to the next stage for continual optimization*. Let $f$ be the target function, and $\theta$ be the parameters. The proposed goal is to find the minimum

$$\min_{\theta} f(\theta). \tag{1}$$

Instead of directly optimizing $f$, we propose to find a series of objective functions $\{f_1(\theta_1), f_2(\theta_2), \cdots, f_n(\theta_n)\}$ with $f_{i-1}$ being able to dynamically grow into $f_i$ and $\theta_{i-1}$ being able to transfer to $\theta_i$, satisfying $f(\theta) = f_n(\theta_n)$. Typically, the functional forms of $\{f_i\}$ are of increasing complexity, and the parameter space is also non-decreasing: $size(\theta_1) \leq size(\theta_2) \leq \cdots \leq size(\theta_n)$. Then, the proposed scheme can be formulated as

$$\hat{\theta}_1 = \operatorname{argmin}_{\theta_1} f_1(\theta_1), \tag{2}$$

$$\hat{\theta}_2 = \operatorname{argmin}_{\theta_2} f_2(\theta_2 | \hat{\theta}_1), \tag{3}$$

$$\cdots$$

$$\hat{\theta}_n = \operatorname{argmin}_{\theta_n} f_n(\theta_n | \hat{\theta}_{n-1}). \tag{4}$$

For deep neural networks, such a network iterative learning process is achieved by network architecture iteration and weight knowledge transferring. In this case, the objective function $f$ is a very deep neural network, and the functional series $\{f_i\}$ is the set of intermediate neural networks. In this research, we design a DynamicNet architecture to allow the network architecture to iteratively evolve and elaborate the application of the proposed learning scheme on two classic networks, i.e., PlainNet and ResNet (He et al., 2015a). We also adopt a modified version of network morphism (Wei et al., 2016) to transfer the knowledge from $\hat{\theta}_{i-1}$ to $\theta_i$ and to facilitate exponential order growth, followed by further optimization to obtain a better-performing $\hat{\theta}_i$.

## 3.2  THE DYNAMICNET ARCHITECTURE

For the proposed learning scheme, the first requirement is a network architecture that allows to dynamically change. Inspired by the observation that modern networks are designed by splitting its body into multiple stages (Simonyan & Zisserman, 2014; He et al., 2015a), we propose a unified architecture called DynamicNet to enable network iterative learning. The proposed architecture follows a modularized architecture design. In this architecture, we calculate the features at several levels from lower ones to higher ones, and each level is further constructed with a repeatable template block to facilitate the network growing.

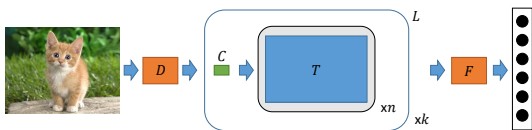

Figure 3: The DynamicNet architecture. A network in the DynamicNet family will allow for architecture change. $D$: fast down-sampling block, $C$: channel switching block, $T$: template block, $L$: level block, $F$: fully connected layer block; $n$: number of blocks, $k$: number of levels.

As illustrated in Fig. 3, the main body of a DynamicNet is composed of $k$ level blocks $L$, in which each level block $L$ further consists of a channel switching block $C$ and several template blocks $T$. The channel switching block $C$ is introduced to simplify the morphing by requiring that layers within the same level block shall have the same channel sizes. This block usually consists of a single convolutional layer[2]. The template block $T$ actually defines the basic module of DynamicNet. Representative network architectures, including PlainNet and ResNet, can be adapted as template blocks in DynamicNet. In addition to the main body, there is also an optional fast down-sampling

---

[2]In this work, unless specially annotated, a convolutional layer will be followed by a BatchNorm layer and a ReLU layer.

block $D$, which is designed to reduce the computational cost. This block typically consists of one or several convolutional layers and pooling layers. There is also a tail $F$ in DynamicNet to represent a classifier connecting to the object categories. It usually consists of a global pooling layer followed by a fully connected layer and a softmax layer.

The DynamicNet is able to represent a large family of networks with modularized architectures. In this work, we verify the proposed learning scheme on two representative network architectures, i.e., PlainNet and ResNet (He et al., 2015a), as shown in Fig. 4. The template block $T$ of a PlainNet is simply composed of a single convolutional layer (Fig. 4(a)). Since the channel switching block has the same architecture as the template block, we shall consider both of them contributing to the number of blocks $n$. An expanded version of the network architecture is illustrated in Fig. 5.

For ResNet (He et al., 2015a), the template block $T$ is a two-branch subnet, with the first branch consisting of two sequential convolutional layers and the second branch a scaled identity mapping layer (residual ratio $r$). These two branches are joined by addition (Fig. 4(b)). In order to be compatible with the ResNet proposed in (He et al., 2015a), we make the architecture of the channel switching block $C$ the same as that of the template block $T$, with the identity mapping layer being replaced with a $1 \times 1$ convolutional layer. The channel switching block shall also be considered for contributing to the number of blocks $n$. We made certain modifications to the ResNet to make it fit for the DynamicNet architecture. In (He et al., 2015a), ResNet is proposed with identity connections, while in this research, we extend them as scaled identity connections controlled by a residual ratio $r$. This extension is to avoid filling the decomposed convolutional filters with all zeros.

Networks in the DynamicNet family allow for exponentially growing their network depths. In general, this can be achieved by making a multiplication on $n$, the number of blocks.

## 3.3  NEURAL NETWORK ARCHITECTURE ITERATION

We explain the detailed network iteration processes for PlainNet and ResNet in the DynamicNet family. We adopt network morphism (Wei et al., 2016) operations as the basic operations in these iteration processes, to guarantee that the network function is unchanged after morphing. Here we focus on the architecture iteration of networks, leaving detailed morphing algorithms to the next section.

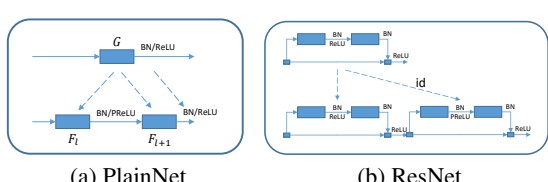

(a) PlainNet       (b) ResNet

Figure 4: Network iteration processes for PlainNet and ResNet in the DynamicNet family.

**The Iteration of PlainNet**

For PlainNet, as shown in Fig. 4(a) and Fig. 5, a convolutional layer $G$ is decomposed into two convolutional layers $F_l$ and $F_{l+1}$, that satisfy the following network morphism equation:

$$\tilde{G}(c_j, c_i) = \sum_{c_l} F_l(c_l, c_i) * F_{l+1}(c_j, c_l) \triangleq F_l \otimes F_{l+1}, \tag{5}$$

where $G$, $F_l$, and $F_{l+1}$ are the convolutional filters associated with the convolutional layers with shapes of $(C_i, C_j, K, K)$, $(C_i, C_l, K_l, K_l)$, and $(C_l, C_j, K_{l+1}, K_{l+1})$. $\tilde{G}$ is a zero-padded version of $G$ whose kernel size is $\tilde{K} = K_l + K_{l+1} - 1$. In the proposed morphing operation for PlainNet, the original BatchNorm and ReLU layers are first duplicated to the end of the second morphed convolutional layer, and the network function will remain the same. However, we still have to insert another BatchNorm/ReLU pair after the first convolutional layer. Here, we assume that such operations are possible. In Section 3.4, we will introduce how to solve the network morphism equation (5) and how to insert a BatchNorm/ReLU pair both with the network function unchanged.

**The Iteration of ResNet**

The iteration process for ResNet is illustrated in Fig. 4(b). As shown, a copy of the original template block is first made. For the second template block, we maintain $r$Id for the shortcut connection, and decompose $(1-r)$Id into two convolutional layers as we did for PlainNet, where $r$ is the residual ratio and Id is the identity mapping. It is worth noting that we have extended the identity connection in (He et al., 2015a) to a scaled one controlled by a residual ratio $r$ in this research. The network function shall not change in the proposed iteration process. As for the BatchNorm and ReLU layers, we simply insert them where they are necessary.

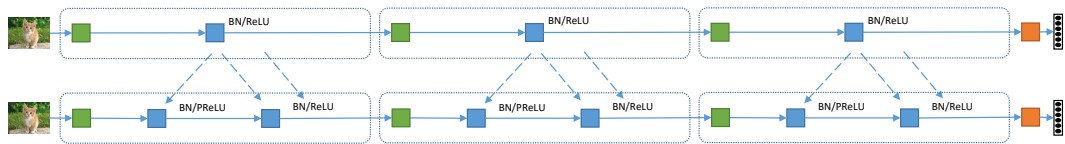

Figure 5: Network iteration process for PlainNet from 7-layer to 10-layer.

**Algorithm 1** The optGD Algorithm for Network Morphism

---

**Input:** $G$ of shape $(C_i, C_j, K, K)$; $C_l, K_l, K_{l+1}$
**Output:** $F_l$ of shape $(C_i, C_l, K_l, K_l)$, $F_{l+1}$ of shape $(C_l, C_j, K_{l+1}, K_{l+1})$
Initialize $F_l$ with unit random noise and initialize $F_{l+1}$ with the initializer described in (He et al., 2015b).
Normalize $G$ with unit standard variance, expand $G$ to $\tilde{G}$ with kernel size $\tilde{K} = K_l + K_{l+1} - 1$ by padding zeros. Keep a record of the standard variance for $\tilde{G}$.
**repeat**
    Fix $F_{l+1}$, and calculate the optimal learning rate for $F_l$ as $\eta_{opt}^l$.
        Update $F_l \leftarrow F_l - \eta_{opt}^l \frac{\partial l}{\partial F_l}$
    Fix $F_l$, and calculate the optimal learning rate for $F_{l+1}$ as $\eta_{opt}^{l+1}$.
        Update $F_{l+1} \leftarrow F_{l+1} - \eta_{opt}^{l+1} \frac{\partial l}{\partial F_{l+1}}$
    Calculate the loss $l = \|\tilde{G} - conv(F_l, F_{l+1})\|^2$
**until** $l = 0$ or $maxIter$ is reached
Multiply $F_l$ with the standard variance of $\tilde{G}$ recorded, and normalize $F_l$ and $F_{l+1}$ with equal standard variances.

**procedure** CALCOPTIMALLEARNINGRATE($G, F_l, F_{l+1}, \frac{\partial l}{\partial F_l}$)
    **for** $\eta = 0, \alpha, 2\alpha$ **do**
        Set $\hat{F}_l \leftarrow F_l - \eta \frac{\partial l}{\partial F_l}$, and calculate the loss $l = \|\tilde{G} - conv(\hat{F}_l, F_{l+1})\|^2$
    **end for**
    Let $l_0, l_1, l_2$ be the losses calculated in the above loop.
    **return** $\frac{\alpha}{2} - \frac{l_1 - l_0}{l_2 - 2l_1 + l_0}$
**end procedure**

---

### 3.4 MORPHING ALGORITHMS

In this section, we shall describe how to decompose one convolutional layer into two convolutional layers that satisfy the network morphism equation (5), and also how to safely insert BatchNorm and ReLU layers into the network so that its network function is unchanged.

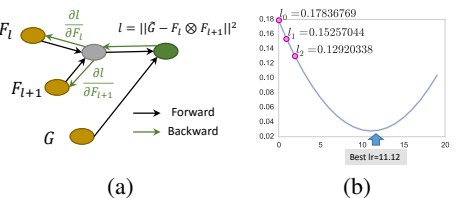

(a)                    (b)

Figure 6: The proposed optGD algorithm for network morphing. (a) Computational graph for the morphism equation (5). (b) Calculation of the optimal learning rate.

**The Morphing of Convolutional Layers**

In the proposed experiment for PlainNet, the algorithm in (Wei et al., 2016) suffered from a dramatic performance drop for the first few iterations. This may be due to its lack of ability to evenly distribute the convolutional filter information into the morphed filters. Consider the case of decomposing one $1 \times 1$ convolutional filter into two $1 \times 1$ convolutional filters, least square based algorithm (Wei et al., 2016) will transfer all information of the original filter into only one of the morphed filters.

To solve this problem, we present a novel and more robust algorithm. Using this algorithm, there was no noticeable performance drop except for the occasions caused by a change of learning rate in the proposed experiments (Fig. 2(e)). The proposed algorithm is based on modified gradient descent. The major difference is that the optimal learning rate for each operand is solved for in the proposed algorithm. Hence, this algorithm is called optimal Gradient Descent (optGD) algorithm in this work. This can be achieved by leveraging the fact that the loss function $l = l(\eta)$ is a quadratic function of the learning rate $\eta$:

$$l(\eta) = \|\hat{F}_l \otimes F_{l+1} - \tilde{G}\|^2 = \|(F_l - \eta \frac{\partial l}{\partial F_l}) \otimes F_{l+1} - \tilde{G}\|^2, \tag{6}$$

where $\hat{F}_l$ is the updated version of $F_l$. Therefore, we can analytically solve for the optimal learning rate $\eta_{opt}$ with any three sample points on the curve.

The algorithm is illustrated in Algorithm 1 and Fig. 6. As shown, we iteratively optimize $F_l$ or $F_{l+1}$ with the other one fixed. The gradients $\frac{\partial l}{\partial F_l}$ and $\frac{\partial l}{\partial F_{l+1}}$ are calculated by the chain-rule, which is implemented in typical deep learning libraries (Jia et al., 2014; Chen et al., 2015b). Its computational graph is also illustrated in Fig. 6(a). For the optimal learning rate, it is calculated by the procedure CALCOPTIMIALLEARNINGRATE in Algorithm 1. We sample three points in the curve $l(\eta)$ with a step size of $\alpha$, and the optimal learning rate can be given by

$$\eta_{opt} = \frac{\alpha}{2} - \frac{l_1 - l_0}{l_2 - 2l_1 + l_0}, \tag{7}$$

where $l_0$, $l_1$, $l_2$ are the losses of $l(\eta)$ when $\eta$ takes values of $0$, $\alpha$, $2\alpha$. Since the minimum of a quadratic function actually does not depend on the three points sampled, we simply set $\alpha = 1$. Fig. 6(b) illustrates an example for the decomposition of a random convolutional filter with $l_0 = 0.17836769$, $l_1 = 0.15257044$, $l_2 = 0.12920338$. The optimal learning rate is calculated as $\eta_{opt} = 11.12$ from Eqn. (7). Note that in the standard SGD algorithm with learning rate 0.1 and mini-batch size 256, its actual learning rate is $0.1/256 \approx 4.0e^{-4}$. 11.12 is much larger than $4.0e^{-4}$ in scale, hence the proposed optGD algorithm converges significantly faster than the standard SGD algorithm.

Eqn. (7) is derived based on the fact that $l(\eta)$ is a quadratic function of $\eta$ (Eqn. (6)). When this condition does not hold, i.e. $l(\eta)$ is an arbitrary continuous function of $\eta$, Eqn. (7) can still give a second-order approximation of the optimal learning rate. In the future, we wish to apply the proposed algorithm to general optimization problems.

**The Morphing of ReLU and BatchNorm**

For ReLU and BatchNorm layers, when they are inserted into a network, the network function shall actually change. The proposed solution is to reduce them to function as an identity mapping layer, so that they can be freely added to the neural network.

In this research, we adopt the solution proposed in (Wei et al., 2016) for the ReLU layer. For any nonlinear activation function $\varphi$, its parametric form $P\text{-}\varphi$ is defined to be any continuous function family that is able to connect $\varphi$ and $\varphi_{id}$, where $\varphi_{id}$ is identity mapping. Its canonical form is defined as

$$P\text{-}\varphi = \{\varphi^a\}|_{a \in [0,1]} = \{(1-a) \cdot \varphi + a\varphi_{id}\}|_{a \in [0,1]}. \tag{8}$$

When $\varphi$ is the ReLU function, $P\text{-}\varphi$ will be the PReLU function (He et al., 2015b). With $a = 1$ as the initialization, the PReLU layer will become an identity layer. It will function as a non-linear one once $a$ has been learned.

For BatchNorm (Ioffe & Szegedy, 2015) defined by

$$newdata = \frac{data - mean}{\sqrt{var + eps}} \cdot gamma + beta, \tag{9}$$

we simply initialize its gamma to ones, and beta to zeros. Mathematically strictly speaking, the mean subtracted by the BatchNorm layer will incur the network function changed unless it is filled with zeros, yet we are only able to show that the expectation of the mean is zero. However, this small perturbation will not result in a noticeable alteration to the network function, since the output is further normalized by the BatchNorm layer after the second convolution ($F_{l+1}$ in Fig. 4(a)) before passing to the next template block. In the proposed experiments, no accuracy degradation was observed using this initialization strategy.

## 4 EXPERIMENTAL RESULTS

### 4.1 EXPERIMENTAL RESULTS ON THE CIFAR10 DATASET

We first conduct experiments on the CIFAR10 (Krizhevsky & Hinton, 2009) dataset, a benchmark dataset for image classification. It consists of 50,000 training images and 10,000 testing images in 10 object categories.

The network inputs are $32 \times 32$ color images with per-channel mean subtracted. Since this dataset is composed of tiny images, the fast down-sampling block in DynamicNet is not necessary[3]. We adopt

---

[3]For ResNet, one convolutional layer is used to fit for its original architecture proposed in (He et al., 2015a).

| Num Block | Num Layer | FixLearn | IterLearn | Abs. Perf. Improv. | Rel. Perf. Improv. |
|---|---|---|---|---|---|
| 2 | 7 | 11.44 | - | - | - |
| 3 | 10 | 10.04 | 9.83 | 0.21 | 2.1 |
| 5 | 16 | 9.87 | 8.36 | 1.51 | 15.3 |
| 9 | 28 | 9.45 | 7.96 | 1.49 | 15.8 |
| 17 | 52 | 11.54 | 7.43 | 4.2 | 36.4 |
| 33 | 100 | 17.77 | 7.57 | 10.2 | 57.4 |

(a) PlainNet

| Num Block | Num Layer | FixLearn | IterLearn r0.5 | FixLearn r0.5 | IterLearn r0.6 | IterLearn r0.7 | IterLearn r0.8 | FixLearn r0.8 | IterLearn r0.9 |
|---|---|---|---|---|---|---|---|---|---|
| 1 | 8 | 11.42 | - | 11.47 | - | - | - | 11.34 | - |
| 2 | 14 | 9.06 | 8.37 | 9.08 | 8.69 | 8.64 | 9.21 | 8.84 | 8.46 |
| 4 | 26 | 8.75 (20) (He et al., 2015a) | 8.04 | 8.3 | 7.43 | 6.82 | 7.25 | 7.68 | 6.94 |
| 8 | 50 | 6.97 (56) (He et al., 2015a) | 6.81 | 8.64 | 6.55 | 5.95 | 6.39 | 7.2 | 6.19 |
| 16 | 98 | 6.61 (110) (He et al., 2015a) | 6.05 | 10.89 | 6.06 | 5.78 | 6.13 | 7.36 | 6.14 |

(b) ResNet

Table 1: Experimental results in error rates on the CIFAR10 dataset. Results adopted from (He et al., 2015a) were reported on different network depths, and their layer numbers are parenthesized.

| NumLayer | FixLearn | IterLearn |
|---|---|---|
| 7 | 39.54 | - |
| 10 | 36.28 | 36.08 |
| 16 | 35.08 | 33.86 |
| 28 | 36.83 | 32.08 |
| 52 | 40.29 | 31.63 |
| 100 | 49.83 | 31.55 |

(a) PlainNet

| Num Layer | FixLearn | IterLearn r0.5 | FixLearn r0.5 | IterLearn r0.8 | FixLearn r0.8 |
|---|---|---|---|---|---|
| 8 | 38.02 | - | 38.86 | - | 38.89 |
| 14 | 34.37 | 33.69 | 34.63 | 33.7 | 34.25 |
| 26 | 32.84 | 31.81 | 32.07 | 31.95 | 31.52 |
| 50 | 31.83 | 29.55 | 36.61 | 29.91 | 31.14 |
| 98 | 28.51 | 28.2 | 44.87 | 27.59 | 30.33 |

(b) ResNet

Table 2: Experimental results in error rates on the CIFAR100 dataset.

a uniform number of levels $k = 3$ for both networks. For PlainNet and ResNet, the depths are $3n + 1$ and $6n + 2$, respectively, where $n$ represents the number of blocks in the level block $L$. The main body of the network architecture is computed on channel sizes of {16,32,64} and feature map sizes of {32,16,8}. This setup is the same as in (He et al., 2015a).

The neural networks were trained using SGD with a batch size of 256 for 320 epochs. The learning rate starts from 0.1 and decreases by a factor of 10 at 160/240 epochs. We use a decay of 0.0001 and a momentum of 0.9. The images are padded with 4 pixels on each side for augmentation, and a random crop and a random horizontal flip are applied.

The first proposed experiment is on PlainNet. In the iteration process, the number of blocks $n$ takes values of $1 + 2^{i-1}$, where 1 is for channel switching and $i$ represents the $i$-th iteration from 1 to 6, resulting in the iteratively evolved network depths of 7, 10, 16, 28, 52, and 100. The experimental results are shown in Table 1(a) and Fig. 2(c). As the results shown, network iterative learning (named as IterLearn) is able to greatly alleviate the degradation problem. The absolute and relative performance improvements can be up to 10.2% and 57.4%. This performance gain is also illustrated by the shadowed purple-pink area in Fig. 2(c). In Fig. 2(c), we also draw the performances of the best-performing models trained with different learning schemes (dashed lines). The global performance improvement is indicated by the gap in between the dashed lines. Besides, Fig. 2(a) and (b) also show the training and testing errors for traditional learning with a fixed architecture (named as FixLearn) and for the proposed learning scheme, where the deeper network (52-layer) could achieve lower error rates in both training and testing than the shallower one (28-layer). This trend in training error reduction is not quite obvious as the errors are actually less than 0.3% and may not be differentiable. This trend can be more clearly observed on the CIFAR100 dataset as illustrated in Fig. 7.

Then, we conduct experiments on ResNet. In the iteration process, the number of blocks $n$ takes values of $2^i$, where $i$ represents the $i$-th iteration from 1 to 5, resulting in the iteratively evolved network depths of 8, 14, 26, 50, and 98. Fig. 2(e) illustrates this iteration in the proposed learning scheme, where the network performance is continually improved. Table 1(b) shows the experimental results. Due to space limitations, we did not show the absolute and relative performance improvements. For the 98-layer ResNet with residual ratio 0.7, the absolute and relative performance improvements can be up to 0.83% and 12.6%.

ResNets with residual ratios $< 1$ are actually not good-performing network architectures. As shown in Table 1(b), when the network depth is increased from 50 to 98, by learning with a fixed architecture, the performance degradation is obvious with residual ratio 0.5 or 0.8. This phenomenon was also observed in (He et al., 2016) (Table 1 and Fig. 2(b)). However, when these networks are learned by the proposed learning scheme, such performance degradation can be avoided. The comparison results between learning with a fixed architecture and the proposed learning scheme is also illustrated in Fig. 2(d) with residual ratio $r = 0.5$, in which the significant performance gain is illustrated as the purple-pink area. Combining with the effectiveness on resolving for the degradation problem on PlainNet, we can conclude that the proposed learning scheme is able to fix the inherent inferiority in these poor-performing network architectures.

Therefore, the effectiveness of network iterative learning has been shown on both representative networks. We observe consistent performance improvements of network iterative learning over learning with a fixed architecture. The degradation problem is also greatly alleviated by the proposed network iterative learning. We believe the reason is that network iterative learning allows full knowledge of learning to be inherited as the initialization when the network is iteratively evolved to a deeper one. Such an iteration of network is regularized, and shall have lower probability to be trapped into a high cost local minimum than conventional scheme of learning with a fixed architecture.

## 4.2 EXPERIMENTAL RESULTS ON THE CIFAR100 DATASET

We also verify the proposed learning scheme on the CIFAR100 dataset. It consists of 50,000 training images and 10,000 testing images in 100 object categories. We follow the same setup and network architectures for Plain-Net and ResNet as in the previous experiments. The results are shown in Table 2, from which we can reach the same conclusions as in the CIFAR10 dataset. For PlainNet and ResNet, the absolute performance improvements can be up to 18.28%, 1.53% respec-

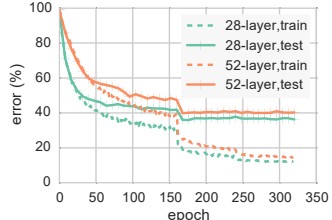 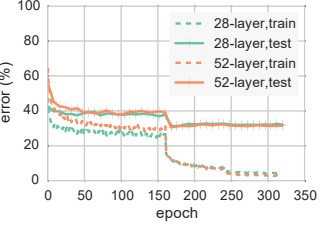

(a) Learn with a fixed architecture.  (b) Network iterative learning.

Figure 7: Network iterative learning vs. learning with a fixed architecture for PlainNet on CIFAR100.

tively, and the relative performance can be up to 36.7%, 5.1%, respectively, for the networks around 100 layers. Due to space limitation, the details of the performance improvements are not included.

In addition to the consistent performance improvement, the degradation problem is also significantly alleviated by the proposed learning scheme for this dataset. From Table 2 we can see that, when the networks are extended from around 50 layers to 100 layers, the proposed learning scheme can achieve a consistent performance improvement for all three networks. Training with a fixed architecture cannot achieve this. Fig. 7(a) and (b) also compare the training and testing errors on PlainNet between these two learning schemes. For the proposed learning scheme, the deeper network (52-layer) could achieve lower error rates in both training and testing than the shallower one (28-layer), while learning with a fixed architecture fails to achieve lower error rates.

## 4.3 EXPERIMENTAL RESULTS ON THE IMAGENET DATASET

We further conduct experiments on ImageNet (Russakovsky et al., 2014). This dataset is composed of 1,000 object categories, with 1.28 million training images and 50,000 validation images. For PlainNet, we set the number of levels to 4 and hence its depth is $4n + 2$. $n$ represents the number of blocks and takes values of $1 + 2^{i-1}$ with $i$ represents the $i$-th iteration from 1 to 5, resulting in the iteratively evolved network depths of 10, 14, 22, 38, 70. The inputs to the network are $224 \times 224$ color patches cropped from resized images with shorter sides randomly sampled in [256,480]. The networks were trained with a batch size of 256 for 100 epochs. The learning

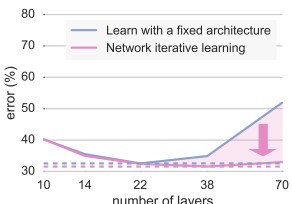

Figure 8: Network iterative learning vs. learning with a fixed architecture for PlainNet on ImageNet.

|  | FixLearn | IterLearn full | IterLearn half |
|---|---|---|---|
| Training Time | 1x | 2x | 1x |
| Error Rate (%) | 17.77 | 7.57 | 7.54 |

Table 3: Training acceleration for the proposed network iterative learning scheme (100-layer PlainNet on CIFAR10).

|  | FixLearn | IterLearn optGD | IterLearn LSQ |
|---|---|---|---|
| Error Rate (%) | 17.77 | 7.57 | 12.55 |

Table 4: Comparison between optGD and LSQ-based (Wei et al., 2016) algorithms solving for the network morphism equation (5) for the proposed network iterative learning scheme (100-layer PlainNet on CIFAR10).

rate starts from 0.1 and decreases with a factor of 10 every 30 epochs.
We use a decay of 0.0001 and a momentum of 0.9. The setup is the same as in (He et al., 2015a).

Fig. 8 illustrates the experimental results. We observe the degradation problem as marked by the blue curve. However, this problem can be significantly alleviated by the proposed learning scheme. The performance gain brought by network iterative learning is indicated by the purple-pink region. Due to time and computational resource constraints, we have not yet carried out experiments for ResNet, which will be completed soon.

## 4.4 EFFICIENCY AND TRAINING ACCELERATION

The proposed learning scheme contains two parts in each iteration period: network iteration (including knowledge transferring) and continual learning. Since the time cost of the first part takes only from seconds to minutes and could be neglected, the time cost of network iterative learning mainly depends on the second part. Due to historical reasons, in this research, we adopted a uniform training strategy for the continual learning and learning with a fixed architecture for 320 epochs. Assume that the training time cost is proportional to the network depth given fixed training steps, one may conclude that the time cost of network iterative learning is about twice the time cost of training the deepest neural network alone.

However, The network iterative learning process is capable of learning new knowledge based on the established knowledge base. Hence, the training process can be greatly accelerated because it avoids re-learning from scratch when the network is going deeper. Table 3 illustrates the experimental results of the training acceleration strategy for the proposed network iterative learning scheme. In this table, "full" means training for 320 epochs and "half" means training for 160 epochs. As shown, the "half" strategy does not incur any performance drop comparing against the "full" training strategy.

## 4.5 COMPARING AGAINST TRADITIONAL OPTIMIZATION ALGORITHMS

We compare the proposed network iterative learning scheme against traditional optimization algorithms. Fig. 9 illustrates the experimental results for the 52-layer and 100-layer PlainNets on the CIFAR10 dataset. As shown, traditional optimization algorithms, including stochastic gradient descent (SGD), Nesterov's accelerated gradient descent (NAG) (Nesterov, 1983; 2013) and Adam (Kingma & Ba, 2014), usually converge to similar performances for a neural network.

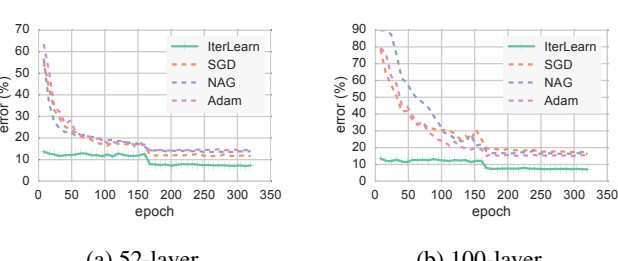

(a) 52-layer        (b) 100-layer

Figure 9: Network iterative learning v.s. traditional optimization algorithms.

While the proposed network iterative learning scheme is able to further reduce the gap between the local minima and the global minimum. This indicates that the problem of local minima with high cost in optimization is common and arises naturally for deep neural networks. The effectiveness of the proposed network iterative learning scheme is also demonstrated.

We also conduct experiments for traditional optimization algorithms including RMSprop (Tieleman & Hinton, 2012), AdaGrad (Duchi et al., 2011) and AdaDelta (Zeiler, 2012). However, they converge to a much worse-performing local minima in the proposed experiment case and their curves are not shown.

### 4.6 The Efficiency of the Proposed optGD Algorithm

Finally, we compare the proposed optGD algorithm with the least square (LSQ) based algorithms proposed in (Wei et al., 2016) for solving the network morphism equation (5). The advantages of the proposed optGD over LSQ when adopted for network morphism of the proposed network iterative learning scheme are mainly in three folds.

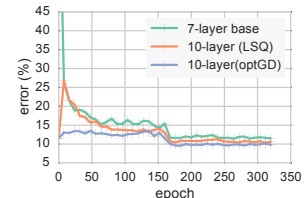

Figure 10: The proposed optGD algorithm v.s. the LSQ-based algorithm (Wei et al., 2016) solving for the network morphism equation.

First, in the proposed experiments, the primary morphing operations are to decompose a $3 \times 3$ convolutional kernel into two $3 \times 3$ convolutional kernels. The proposed optGD algorithm is able to do this directly. While for LSQ in (Wei et al., 2016), one has to decompose a $3 \times 3$ convolutional kernel into a $3 \times 3$ convolutional kernel and a $1 \times 1$ convolutional kernel, and then expand the latter to a $3 \times 3$ convolutional kernel in order to achieve the desired morphing.

Second, the proposed optGD algorithm does not incur an additional sharp error increase at the start besides the learning rate change (Fig. 10 blue curve). While in (Wei et al., 2016), there is an additional sharp error increase at the start besides the learning rate change (Fig. 10 orange curve). Such sharp error increase happened when we were expanding the $1 \times 1$ convolutional kernel into a $3 \times 3$ one. It should be caused by filling the convolutional filters with too many zeros, as the Net2Net approach in (Chen et al., 2015a) also encounters the same problem due to the large amount of zeros in identity layers added into the network (Fig. 5 in (Chen et al., 2015a)).

Third, the proposed optGD algorithm is much more robust than the LSQ-based algorithm in (Wei et al., 2016). This is probably because the proposed optGD algorithm is able to evenly distribute the convolutional filter information into the morphed filters. While LSQ-based algorithm will likely transfer all information of the original filter into only one of the morphed filters. Table 4 compares the experimental results for PlainNet on the CIFAR10 dataset. As can be seen, the proposed optGD algorithm is much more effective than the LSQ-based algorithm (7.57% vs 12.55%) solving the network morphism equation for the proposed network iterative learning scheme. It is also worth noting that, even when combined with the LSQ algorithm, the proposed scheme still outperforms the learning with a fixed architecture scheme by a large margin (12.55% vs 17.77%).

## 5 Conclusions

In this research, we show that the problem of local minima with high cost in optimization is common and arises naturally for deep neural networks. In an attempt to further reduce the gap between the local minima and the global minimum, we present a novel learning scheme called network iterative learning for deep neural networks. Different from traditional optimization algorithms that usually optimize directly on a static objective function, the proposed network iterative learning scheme contains a novel deep dynamic neural network architecture. Extensive experiments have been carried out to demonstrate the effectiveness of the proposed network iterative learning scheme.

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
