# OpenReview forum: "Network Iterative Learning for Dynamic Deep Neural Networks via Morphism"
_ICLR.cc/2018/Conference — Reject_

### Official Review · AnonReviewer2 · 2017-11-27
**The submission looks interesting and is well-written**

**Rating:** 7
**Confidence:** 4

**Review:**

This submission develops a learning scheme for training deep neural networks with adoption of network morphism (Wei et al., 2016), which optimizes a dynamic objective function in an iterative fashion capable of adapting its function form when being optimized, instead of directly optimizing a static objective function. Overall, the idea looks interesting and the manuscript is well-written. The shown experimental results should be able to validate the effectiveness of the learning scheme to some extent.

It would be more convincing to include the performance evaluation of the learning scheme in some representative applications, since the optimality of the training objective function is not necessarily the same as that of the trained network in the application of interest.

Below are two minor issues:

- In page 2, it is stated that Fig. 2(e) illustrates the idea of the proposed network iterative learning scheme for deep neural networks based on network morphism. However, the idea seems not clear from Fig. 2(e).

- In page 4, “such network iterative learning process” should be “such a network iterative learning process”.

---

### Official Review · AnonReviewer1 · 2017-11-27
**Interesting discussion, but not novel enough**

**Rating:** 5
**Confidence:** 2

**Review:**

This paper proposes an iterative approach to train deep neural networks based on morphism of the network structure into more complex ones. The ideas are rather simple, but could be potentially important for improving the performance of the networks. On the other hand, it seems that an important part of the work has already been done before (in particular Wei et al. 2016), and that the differences from there are very ad-hoc and intuition for why they work is not present. Instead, the paper justifies its approach by arguing that the experimental results are good. Personally, I am skeptical with that, because interesting ideas with great added value usually have some cool intuition behind them. The paper is easy to read, and there does not seem to exist major errors. Because I am not an active researcher in the topic, I cannot judge if the benefits that are shown in the experiments are enough for publication (the theoretical part is not the strongest of the paper).

---

### Official Review · AnonReviewer4 · 2017-12-04
**Interesting paper, but requires more clarifications on novelty and experiment results**

**Rating:** 5
**Confidence:** 3

**Review:**

This paper proposed an iterative learning scheme to train a very deep convolutional neural network. Instead of learning a deep network from scratch, the authors proposed to gradually increase the depth of the network while transferring the knowledge obtained from the shallower network by applying network morphism.

Overall, the paper is clearly written and the proposed ideas are interesting. However, many parts of the ideas discussed in the paper (Section 3.3) are already investigated in Wei et al., 2016, which limits the novel contribution of the paper. Besides, the best performances obtained by the proposed method are generally much lower than the ones reported by the existing methods (e.g. He et al., 2016) except cifar-10 experiment, which makes it hard for the readers to convince that the proposed method is superior than the existing ones. More thorough discussions are required.

---

### Decision · Program_Chairs · 2018-01-29
**ICLR 2018 Conference Acceptance Decision**

**Decision:**

Reject

**Comment:**

The paper presents a variant of network morphism (Wei et al., 2016) for dynamically growing deep neural networks. There are some novel contributions (such as OptGD for finding a morphism given the parent network layer). However, in the current form, the experiments mostly focus on comparisons against fixed network structure (but this doesn't seem like a strong baseline, given Wei et al.'s work), so the paper should provide more comparisons against Wei et al. (2016) to highlight the contribution of this work. In addition, the results will be more convincing if the state-of-the-art performance can be demonstrated for large-scale problems (such as ImageNet classification).